# Reasoning with Transformer-based Models:
# Deep Learning, but Shallow Reasoning

**Chadi Helwe**                                         CHADI.HELWE@TELECOM-PARIS.FR
**Chloé Clavel**                                       CHLOE.CLAVEL@TELECOM-PARIS.FR
**Fabian Suchanek**                                       SUCHANEK@TELECOM-PARIS.FR
*Télécom Paris, Institut Polytechnique de Paris, France*

## Abstract

Recent years have seen impressive performance of transformer-based models on different natural language processing tasks. However, it is not clear to what degree the transformers can reason on natural language. To shed light on this question, this survey paper discusses the performance of transformers on different reasoning tasks, including mathematical reasoning, commonsense reasoning, and logical reasoning. We point out successes and limitations, of both empirical and theoretical nature.

## 1. Introduction

In recent years, language models have achieved impressive results on a variety of natural language processing (NLP) tasks, such as recognizing textual entailment, machine reading comprehension, and machine translation. Most of these language models are based on variants of the transformer architecture [Vaswani et al., 2017], for example BERT [Devlin et al., 2019], T5 [Raffel et al., 2020a], and GPT-3 [Brown et al., 2020]. These models depend entirely on the attention mechanism, and thus eliminate the need for recurrent computations used by LSTMs [Hochreiter and Schmidhuber, 1997] and GRUs [Cho et al., 2014]. They can easily learn long-range dependencies, and the computation can be parallelized efficiently. Today's models contain millions or even billions of parameters. The models are pre-trained on large unlabeled corpora, and then later fine-tuned to tackle a specific NLP task. For example, the pre-trained BERT can reply to questions such as the following:

|  |  |
|---:|:---|
| **Context:** | The iPhone is produced by [MASK]. |
| **Expected answer:** | Apple |
| **Model answer:** | Apple |

However, this performance is deceiving: If we introduce a trap word, the pre-trained BERT model replies completely differently:

|  |  |
|---:|:---|
| **Context:** | Samsung. The iPhone is produced by [MASK]. |
| **Expected answer:** | Apple |
| **Model answer:** | Samsung |

Here, the BERT model got distracted by the additional word (a technique called *mispriming* [Kassner and Schütze, 2020]). Thus, the question arises to what degree such models really "understand" the natural language text, and to what degree they merely respond to statistical cues. This question is of utmost importance, because if we start relying on such

language models, there is the danger that we obtain good responses only in common test settings, and completely abstruse replies in less common settings.

In this survey paper, we shed light on this question by investigating some of the most complex natural language tasks: those that involve reasoning. That is, we look at test data sets that have explicitly been designed to test the limitations of transformer-based models, and we investigate to what degree the models really "understand" these tasks. While several survey papers have focused on transformer-based models and on their applications [Rogers et al., 2020b, Qiu et al., 2020, Xia et al., 2020, Yates et al., 2021], the capabilities of transformer-based models in reasoning tasks have so far not been surveyed. Our paper is organized as follows. In Section 2, we describe some common pitfalls that all models need to handle in order to reason on natural language text. Section 3 analyzes the performance of transformer-based models on different reasoning tasks. In Section 4, we describe the theoretical limitations of the transformer architecture, and show, in Sections 4.2-4.3, that they impact natural language reasoning. We conclude in Section 5. The appendix contains a detailed list of basic models (Appendix A) and challenging datasets (Appendix B), as well as of the model performances (Appendix C).

## 2. Common Pitfalls for BERT-like Models

We discuss here some common pitfalls that any approach needs to handle in order to reason on natural language. Our discussion focuses on BERT, but the phenomena may affect other transformer-based models as well.

### 2.1 Negation

The pre-trained BERT model cannot differentiate between positive and negative statements. As an example, take this sentence from the LAnguage Model Analysis (LAMA) dataset [Petroni et al., 2019], where BERT performs well:

|  |  |
|---:|:---|
| **Context:** | Marcel Oopa died in the city of [MASK]. |
| **Expected answer:** | Paris |
| **Model answer:** | Paris (-2.3), Lausanne (-3.3), Brussels (-3.3) |

When Kassner and Schütze [2020] added the word "not", BERT delivered the exact same top-ranked result:

|  |  |
|---:|:---|
| **Context:** | Marcel Oopa did not die in the city of [MASK]. |
| **Expected answer:** | any city different from Paris |
| **Model answer:** | Paris (-2.4), Helsinki (-3.5),Warsaw (-3.5) |

This phenomenon was also confirmed by Ettinger [2020]. Kassner and Schütze [2020] show that BERT can be fine-tuned to pay attention to the negation. Thus, it is essential to add examples with negation to the training set. Niven and Kao [2019] point out that these examples should be diverse enough to not rely only on the word "not", and Hosseini et al. [2021] propose an unlikelihood objective function to learn to differentiate between positive and negative statements.

## 2.2 Mispriming

The ability to distinguish useful from distracting contexts is an essential building block for any reasoning task. We have already seen an example of mispriming in the introduction. Mispriming can, in principle, affect any task, and thus also reasoning in particular.

Interestingly, mispriming works only when the distracting word is of the same type as the expected answer (companies, in our example). The pre-trained BERT is not easily misled by primes of other types [Niven and Kao, 2019]. Misra et al. [2020] also show that the problem of mispriming can be overcome by providing more context. In the following sentence, for example, the mispriming fails:

| | |
|---:|:---|
| **Context:** | Samsung. The iPhone was produced by [MASK], |
| | whose CEO was Steve Jobs |
| **Expected answer:** | Apple |
| **Model answer:** | Apple |

This shows that, although there is some dependency on misprimes, their power decreases when sentences provide more context.

## 2.3 Pattern Heuristics

Fine-tuned BERT models have a tendency to learn simple pattern-based heuristics. For example, BERT can be trained to perform well on textual entailment in the MNLI dataset [Williams et al., 2018]:

| | |
|---:|:---|
| **Premise:** | The actor and the professor mentioned the lawyer. |
| **Hypothesis:** | The professor mentioned the lawyer. |
| **Expected answer:** | Entailment |
| **Model answer:** | Entailment |

To better understand the performance of BERT, McCoy et al. [2019b] designed the HANS (Heuristic Analysis for NLI Systems) dataset. HANS makes BERT fail as follows:

| | |
|---:|:---|
| **Premise:** | The doctors advised the presidents and the tourists. |
| **Hypothesis:** | The presidents advised the tourists. |
| **Expected answer:** | Non entailment |
| **Model answer:** | Entailment |

This shows that the model learned the "lexical overlap heuristic", which assumes that a premise entails all hypotheses constructed from words in the premise. This problem can be addressed by adding more HANS-like examples to the training dataset.

## 2.4 Word Order

Different studies [Ettinger, 2020, Sankar et al., 2019, Pham et al., 2020, Gupta et al., 2021] have shown that BERT-like models are unperturbed by grammatically incorrect sentences: If presented with a sentence of randomly shuffled words, they will still reply correctly. This insensitivity to order can also mislead textual entailment. For example, the pre-trained BERT fine-tuned on the MNLI dataset fails to provide the correct answer in the following

case [McCoy et al., 2019b,a]:

|  |  |
|---:|:---|
| **Premise:** | The doctor visited the lawyer |
| **Hypothesis:** | The lawyer visited the doctor |
| **Expected answer:** | Non entailment |
| **Model answer:** | Entailment |

This issue can be solved by augmenting the training set with modified word order instances with their respective labels or by fine-tuning the model on sensitive word ordering tasks such as CoLA [Warstadt et al., 2019].

## 3. Types of Reasoning with Transformer-based Models

### 3.1 Horn Rule Reasoning

A rather simple way of logical reasoning is to infer a conclusion from a set of premises and rules. Transformer-based models are able to perform such kind of reasoning [Clark et al., 2020, Talmor et al., 2020, Betz et al., 2020] without any external knowledge, if both the rules and the facts are mentioned explicitly in the text. They can even generate the proofs [Saha et al., 2020, Tafjord et al., 2020, Gontier et al., 2020]. Here is an example from the ParaRules dataset [Clark et al., 2020]:

|  |  |
|---:|:---|
| **Context:** | Fact 1: Erin is young. |
| | Fact 2: Erin is not kind. |
| | Fact 3: Peter is nice. |
| | Rule 1: If someone is young and not kind then they are big. |
| **Question:** | Is Erin big? |
| **Expected answer:** | *Conclusion:* Erin is big. |
| | *Proof:* (Fact 1 & Fact 2) → Rule 1 → Conclusion |

In this task, the best model, a fine-tuned T5-11B, achieves an accuracy above 95% in proof generation and question answering. A transformer-based model can thus solve the task nearly perfectly.

### 3.2 Commonsense Reasoning

Commonsense reasoning is any reasoning task that requires background knowledge that humans commonly have. For example, the instruction "Can you do a Napoleon for the camera?" requires commonsense reasoning to realize that the word "Napoleon" expresses a specific pose [Bender and Koller, 2020]. Several studies have shown that BERT learned a certain amount of commonsense knowledge during pre-training [Petroni et al., 2019, Davison et al., 2019, Bosselut et al., 2019, Zhou et al., 2020b, Cui et al., 2020]. Consider, for example, the LAMA dataset [Petroni et al., 2019], which asks:

|  |  |
|---:|:---|
| **Context:** | Ravens can [MASK] |
| **Expected answer:** | fly |
| **Model answer:** | fly |

The model (the pre-trained BERT-large) is able to recall such commonsense knowledge.

This good performance has prompted the research community to develop datasets that specifically probe the commonsense reasoning of transformer models. Prominent datasets are COSMOS QA [Huang et al., 2019], CommonsenseQA [Talmor et al., 2019], the Winograd Schema Challenge [Levesque et al., 2012], SWAG [Zellers et al., 2018], ReCoRD [Zhang et al., 2018], CODAH [Chen et al., 2019], and PIQA [Bisk et al., 2020]. Transformer-based models can indeed achieve a high performance (often > 75%) on these datasets, but only with additional methods. These include data augmentation techniques [Yang et al., 2020], multi-task learning [Lourie et al., 2021], and fusing knowledge graphs into language models [Xu et al., 2021]. The following is an example from the CommonsenseQA dataset [Talmor et al., 2019]:

|                                    |                                                   |
|-----------------------------------:|:--------------------------------------------------|
| **Question:**                      | Bats have many quirks, with the exception of ?    |
| **Expected Answer:**               | Laying eggs                                       |
| **Model w/o knowledge graph fusing:** | Eating bugs                                     |
| **Model w/ knowledge graph fusing:**  | Laying eggs                                     |

The above example shows that providing the model with information from a knowledge graph helps the model to correctly answer the question. However, several studies [Forbes et al., 2019, Zhou et al., 2020b, Lin et al., 2020, Boratko et al., 2020, Singh et al., 2021] show that when the datasets are specifically changed to target the weaknesses of transformer-based models (for example, by adversarial instances), the models fail. Here is an example from the COM2SENSE dataset [Singh et al., 2021], which asks the model to judge whether a given sentence is logically coherent or not:

|                       |                                                            |
|----------------------:|:-----------------------------------------------------------|
| **Context:**          | Expecting ten fish in the net, Sammy was                   |
|                       | thrilled to see forty fish swimming in there.              |
| **Expected answer:**  | Coherent                                                   |
| **Model answer:**     | Coherent                                                   |

The authors created a counterpart to this question by modifying a few words:

|                       |                                                            |
|----------------------:|:-----------------------------------------------------------|
| **Context:**          | Expecting ten fish in the net, Sammy was                   |
|                       | thrilled to see *five* fish swimming in there.             |
| **Expected answer:**  | Incoherent                                                 |
| **Model answer:**     | Coherent                                                   |

When the model (UnifiedQA-3B [Khashabi et al., 2020], a multi-task trained model) is tricked this way, it fails to predict correctly. This shows that the model can fall prey to relatively simple modifications, and does not really reason.

### 3.3 Event-based Commonsense Reasoning

Some commonsense reasoning tasks are concerned with the usual sequence of events. For example, the TIMEDIAL dataset [Qin et al., 2021] evaluates temporal reasoning capabilities in dialogs. The TORQUE dataset [Ning et al., 2020] asks temporal relation questions such as which events have already finished, given a short passage of text. In a similar spirit, the MCTACO dataset [Zhou et al., 2019] asks:

**Context:** Mr. Barco has refused US troops or advisers but has accepted US military aid.
**Question:** What happened after Mr. Barco accepted the military aid?
**Choices:** (A) the aid was denied, (B) he received the aid, (C) things started to progress

The best model is a fine-tuned BERT model that uses normalization to convert numerical expressions such as "30 months" to "2.5 years". It achieves an F1-score of 69.9% (while human performance has an F1-score of 87.1%). In the same spirit, Zhou et al. [2021] developed TRACIE, a temporal reasoning textual entailment dataset that asks whether one event preceded another one. The authors use distant supervision from Wikipedia, and a symbolic reasoning model called SymTime. This approach predicts the end time of an event by having two transformer models that predict the start time and the duration of this event and symbolically compare them against the prediction of another start time event. With this, the authors achieve an accuracy of about 71% (with variations for different subtasks). Like the "normal" commonsense tasks, event-based tasks can be solved rather well by transformer-based models. However, this works mainly when symbolic machinery (such as date normalization and symbolic reasoning) or background knowledge (such as Wikipedia) is added. Human performance, in the high nineties, remains unachieved.

### 3.4 Implicit Reasoning

We now turn to implicit reasoning tasks, where (different from the tasks in Section 3.1), the rules and facts are not given explicitly. Many of these tasks can be solved by transformer-based models. Here is an example from the SNLI dataset [Bowman et al., 2015]:

**Premise:** Three girls take cover under their umbrellas.
**Hypothesis:** Nobody has umbrellas.
**Expected answer:** Contradiction
**Model answer:** Contradiction

In this task, a RoBERTa-large model, trained with a few-shot learning method [Wang et al., 2021a], achieves an accuracy of 93.1%. However, these datasets contain superficial cues that the models can take advantage of [Schlegel et al., 2020, Huang and Zhu, 2021, Lin et al., 2021]. To adequately evaluate the understanding of a model, several more challenging logical reasoning tasks have been designed, which mostly take the form of machine reading comprehension. LogiQA [Liu et al., 2020b], for example, is a multiple choice dataset translated from the National Civil Servants Examination of China:

**Context:** David knows Mr. Zhang's friend Jack, and Jack knows David's friend Ms. Lin. Everyone of them who knows Jack has a master's degree, and everyone of them who knows Ms. Lin is from Shanghai.
**Question:** Who is from Shanghai and has a master's degree?
**Choices:** (A) David (B) Jack (C) Mr. Zhang (D) Ms. Lin

The best language model is a pre-trained RoBERTa model [Liu et al., 2019] fine-tuned on the training set and has an accuracy of 35.31% (while the best human performance is 96%) [Liu et al., 2020b]. Several other benchmarks in this vein also show bad performance: ReClor [Yu et al., 2020], QuAIL [Rogers et al., 2020a], ConTRoL [Liu et al., 2020a], Strate-

gyQA [Geva et al., 2021], AR-LSAT [Zhong et al., 2021], and CLUTRR [Sinha et al., 2019]. This shows that transformer-based models are currently unable to build a representation of a longer text and draw a logical conclusion from it. This weakness can be remedied to some degree by adding symbolic representations on top of RoBERTa, such as graph-based modules [Huang et al., 2021, Ouyang et al., 2021], or logical information [Wang et al., 2021b]. Other approaches develop neuro-symbolic methods, which teach reasoning strategies by gradient-based optimisation [Minervini et al., 2020], or combine probabilistic logic programming with neural networks [Manhaeve et al., 2018]. Integrating logical information into RoBERTa pushes the performance on the easier questions of ReClor to 81.4%. However, the more difficult questions of these datasets incur performances of 50%-60%. The same is true for comparison-based tasks. The RICA dataset [Zhou et al., 2020a], for example, asks:

| | |
|---:|:---|
| **Context:** | A prindag is smaller than a flurberg, |
| | so a flurberg is [MASK] likely to contain a prindag. |
| **Expected answer:** | more |

Pre-trained and fine-tuned language models such as GPT-2 [Radford et al., 2019] and RoBERTa achieve a dismal performance of 50% on unseen inferences. Thus, these models are unable to learn comparisons between (fictitious) objects.

## 3.5 Mathematical Reasoning

Mathematical reasoning is the process of reasoning about different mathematical aspects such as arithmetic operations, numerical comparison, counting, and sorting. The level of complexity can range from solving simple mathematical equations to proving theorems. The following is an example of a math problem that is not linguistically complex, taken from the DeepMind mathematics dataset [Saxton et al., 2019]:

| | |
|---:|:---|
| **Context:** | Calculate -841880142.544 + 411127 |
| **Expected answer:** | -841469015.544 |

This task can be solved by GPT-3 [Henighan et al., 2020]. Along the same line, Lample and Charton [2019] show that a transformer network can compute function integrals, and solve differential equations. The next more complex problems are math word problems (MWP), which consist of a short text that describes a mathematical problem (such as a one-unknown math problem) and a question. This task requires a model to extract relevant information from the text to perform mathematical reasoning to solve it. The most prominent MWP datasets are MAWPS [Koncel-Kedziorski et al., 2016] and ASDiv-A [Miao et al., 2020] (both for one-unknown arithmetic problems). However, these datasets can be solved by models even when the order of the words is modified and when the questions are omitted, proving that the models rely on heuristic patterns found in the problem narrative. To remove these artifacts, Patel et al. [2021] developed the SVAMP dataset, which applies simple variations to ASDiv-A. The following is an example:

|                   |                                          |
|------------------:|:-----------------------------------------|
| **Context:**      | Jack had 8 pens and Mary had 5 pens.     |
|                   | Mary gave 3 pens to Jack.                |
| **Question:**     | How many pens does Jack have now?        |
| **Expected answer:** | $8 + 3 = 11$                          |

On this dataset, a trained model achieves an accuracy of around 65%. Among the different mathematical operators (+,-,/,*), the model accuracy ranges from 65.3% for divisions to 35.8% for multiplications. Also, the performance drops drastically when the equations have more than two numbers or more than one operator.

An even more complicated dataset is MATH [Hendrycks et al., 2021], which consists of competition problems in mathematics:

|                   |                                          |
|------------------:|:-----------------------------------------|
| **Context:**      | Tom has a red marble, a green marble, a blue marble, |
|                   | and three identical yellow marbles.       |
| **Question:**     | How many different groups of two marbles can Tom choose? |
| **Expected answer:** | There are two cases here: either Tom chooses |
|                   | two yellow marbles (1 result), or he chooses two marbles of different |
|                   | colors (($\frac{4}{2}$)= 6 results). The total number of distinct pairs |
|                   | of marbles Tom can choose is $1 + 6 = 7$  |

Here, the best model, a fine-tuned GPT-2 model, achieves an accuracy of only 6.9%.

Another dataset at the boundary of what is currently feasible is the IsarStep benchmark [Li et al., 2021], which is concerned with mathematical proofs:

|                   |                                          |
|------------------:|:-----------------------------------------|
| **Context:**      | $2b^2 = a^2 \Rightarrow [Missing\ Proposition] \Rightarrow \exists\ c \in\ \mathbb{Z}.\ a = 2c$ |
| **Expected answer:** | a is even                            |

The authors developed a hierarchical transformer model, which outperforms all the other tested baselines with an accuracy of 22.8% for the top-1 prediction, and an accuracy of 35.2% for the top-10 predictions. Other mathematical theorem proving datasets in the same spirit are HOList [Bansal et al., 2019] and MetaMathStep [Polu and Sutskever, 2020]. In conclusion, these tasks show that transformer-based models cannot be trained to "understand" mathematical word problems and to "generate" mathematical proofs. In contrast to simple mathematical problems (as the example we mentioned above), which are not linguistically complex, such challenging tasks require more than huge transformer-based models to achieve high performance.

### 3.6 Summary

In all of these reasoning tasks, transformer-based models rarely achieve human performance. That is not surprising, given that they are general-purpose tools that feed mainly from training data, and lack any symbolic machinery that is commonly considered essential for such tasks. In fact, it is impressive that the models perform so well at all.

Among the different reasoning tasks, we find that when the transformer-based models are explicitly given all the information required to perform deductive reasoning, such as facts and rules, the models can easily learn logical reasoning. However, when this information is stated only implicitly in the text or in the supervision, the models struggle. In

terms of commonsense reasoning, transformer-based models have a certain degree of commonsense knowledge learned during pre-training. However, they can be easily disrupted with adversarial commonsense instances. They are also limited in logical reasoning over events and physical commonsense.

We thus see that the strength of transformer-based models comes from two components: simple patterns in the training data, combined with background knowledge from the pre-training. This combination allows the models to perform well on tasks such as Horn Rule Reasoning (where the model learns a pattern on the training data), simple commonsense reasoning (where the answer was learned from the pretraining), and simple mathematical calculations (where the model learns a pattern during training). However, when these elements are absent, the models struggle. We have seen several commonsense datasets that specifically avoid patterns, or use adversarial patterns. Here, the models fail. In particular, textual understanding remains out of reach for now, if the tasks are sufficiently different from each other to avoid patterns, and if fictional entities are used (for which no background knowledge is available). Mathematical reasoning, too, falls into this category, if the tasks do not follow a pattern.

This does not mean that the tasks would be unsolvable in general: Several studies [Huang et al., 2021, Ouyang et al., 2021, Wang et al., 2021b, Yang et al., 2020, Lourie et al., 2021, Xu et al., 2021] show that the addition of symbolic knowledge (such as date normalization, quasi-logical reasoning, and graph-based modules) and the use of supplementary techniques such as data augmentation, multi-task learning, and knowledge-base fusion improve the performance. Such tools may thus hold the key to address even the harder reasoning problems.

## 4. Impossible Reasoning Tasks

We have seen that transformer-based models can be trained and tuned to work on reasoning tasks, and that some tasks require additional machinery. In this section, we turn to tasks that BERT will never be able to solve without additional machinery, no matter the amount of tuning.

### 4.1 Theoretical Limitations of Transformers

Hahn [2020] studied the theoretical limitations of transformers. The main limitations come from the fact that self-attention does not have the same level of expressiveness as recurrent models such as LSTMs. In particular, transformers cannot emulate a stack and finite-state automata. Based on this insight, Hahn proved that transformer-based networks cannot model two languages, known as Parity and Dyck-2. Parity is the set of bit strings where the number of 1s is even. Dyck-2 is the language of strings that are balanced sequences of round brackets "()" and square brackets "[]". Hahn shows that for any transformer network, we can find an integer $N$ such that strings of these languages longer than $N$ cannot be recognized. That is: to recognize such strings, the number of heads and layers of the model would have to increase with the input length $N$. Bhattamishra et al. [2020b] have verified these results in practice, and showed that when the input length is bounded during training, LSTM can generalize to longer instances, whereas transformer architectures cannot. Other theoretical limitations are studied in Bhattamishra et al. [2020a]. These

limitations are of rather theoretical nature. And yet, they have a very concrete impact on natural language reasoning. To show this, we designed two experiments: the light switch task and the cake task. All our datasets and the code of the experiments can be found at https://github.com/dig-team/FailBERT.

### 4.2 Light Switch Task

Our first task puts the Parity language into practice. The input is a word of the language $((a|b) \text{ and })*(a|b)$, with $a =$ "I ate a pizza" and $b =$ "I operated the light switch", i.e., the input is a sentence that describes a sequence of these two activities. Assuming that the light is off in the beginning, the goal is to determine whether the light is on in the end (which corresponds to an odd number of switches). This is a simple reasoning task that a school child can solve.

> **Context:** I operated the light switch, and I ate a pizza, and I ate a pizza.
> **Expected answer:** ON

We fine-tuned a pre-trained RoBERTa model for 50 iterations on 20k examples, each containing up to 20 $a$'s and $b$'s. On the training and validation datasets, the model achieves an F-score $> 0.99$. However, when testing on examples that contain more than 20 $a$'s and $b$'s, we obtain on average a random precision of 0.50. This confirms that the theoretical limitation of the transformer-based model has practical implications for natural language reasoning.

### 4.3 Cake Task

Our next task puts the Dyck language into practice. The input to the task is a word of the language $S \rightarrow \epsilon|SS|aSa'|bSb'$, where $a =$ "I add a peanut layer to my cake", $a' =$ "I eat a peanut layer from my cake", $b =$ "I add a chocolate layer to my cake", and $b' =$ "I eat a chocolate layer from my cake" (with the conjunction "and" in suitable places). The goal is to determine whether this sequence of steps is possible and the cake is gone. Again, this is a simple reasoning task that a child can solve on a sheet of paper (or with suitable baking tools).

> **Context:** I add a peanut layer and I eat a peanut layer.
> **Expected answer:** Yes

> **Context:** I add a peanut layer and I eat a chocolate layer.
> **Expected answer:** No

We fine-tuned a pre-trained RoBERTa model on 24k examples, each with up to 20 items, and with nesting depths up to 15, for 50 iterations. Again, the model achieves an F-score $> 0.99$ on the training and validation sets. However, when testing on examples that contain more than 20 items, and on depths larger than 15, we obtain, as expected, on average a dismal F-score of 0.55. This shows again that the theoretical limitation of BERT-like models lead to very concrete limitations on natural language reasoning.

## 5. Conclusion

This survey paper has shown that transformer-based models can perform a shallow level of reasoning on textual data but lack deeper reasoning capabilities. The first stumbling stones are some common pitfalls for BERT-like models: word order, negation, shallow patterns, and priming problems. The models have to be explicitly trained to deal with these. We have then discussed several reasoning tasks – from the simple Horn rule reasoning to the more complex commonsense, textual understanding, and mathematical tasks. On these tasks, the performance of transformer-based models is significantly behind human performance. One promising direction of research here is to add symbolical knowledge to the system – an approach that has been pursued with success on some of the tasks. However, we have also recalled that transformer-based models have theoretical limitations in that they cannot model the languages Dyck-2 and Parity. We have shown on small reasoning tasks that these theoretical limitations, too, can hinder reasoning on natural language. Further research could explore how different types of positional encodings (such as learned embeddings, sinusoidal embeddings, or CAPE [Likhomanenko et al., 2021]) and different attention mechanisms (such as saturated attention [Merrill et al., 2021]) could help the models overcome even these limitations.

**Acknowledgements.** This work was partially funded by ANR-20-CHIA-0012-01 ("NoRDF").

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

## Appendix

This appendix describes the basic transformer-based models (Appendix A), the datasets mentioned above that are particularly challenging (Appendix B), and the performances of the models (Appendix C).

## Appendix A. Models

### A.1 The Transformer Model

The transformer model [Vaswani et al., 2017] is a neural network architecture that is based entirely on the attention mechanism. Thereby, it eliminates the need for recurrent computation used by LSTMs [Hochreiter and Schmidhuber, 1997] and GRUs [Cho et al., 2014]. Also, it easily learns long-range dependencies, and it allows the computation to be performed in parallel. The transformer achieved state of the art results in machine translation.

### A.2 BERT

BERT [Devlin et al., 2019] is a pre-trained language mode that consists of a stack of transformer blocks. BERT was pre-trained on two large corpora: The Books Corpus [Zhu et al., 2015] (800M words) and Wikipedia (2500M words). BERT was pre-trained on two tasks: Masked Language Modeling (MLM) and Next Sentence Prediction (NSP).
The task of MLM consists of training the model to predict a masked word given the other words in a sentence. The dataset is constructed by choosing 15% of its tokens to be masked, and by replacing 80% of them with the [MASK] token, 10% with a random token, and 10% with the original token. The BERT model is trained to predict the masked word based on the context of the sentences. The task of NSP consists of training the model to learn the relationship between two sentences by taking as input two sentences and predicting if one sentence follows the other.

The BERT-base model consists of 12 layers of transformer blocks with a hidden size of 768. It has 110M parameters. The BERT-large model consists of 24 layers of transformer blocks with a hidden size of 1024, and has 340M parameters.

### A.3 RoBERTa

RoBERTa [Liu et al., 2019] is an improved BERT model, which achieves better results than BERT on different NLP tasks. The model was pre-trained longer and on a larger dataset than BERT, by including three more datasets, namely the CommonCrawl News dataset of 63 million articles, the Web text corpus, and the Stories Corpus from Common Crawl. The authors pre-trained the model on longer sequences, removed the NSP task, and introduced dynamic masking (a masking technique to dynamically change the masked tokens after each training epoch). Both variants of RoBERTa, RoBERTa-base and RoBERTa-large, have an

architecture that is similar to the one of BERT-base and BERT-large, respectively, but use more parameters.

### A.4 BART

BART [Lewis et al., 2019] is a denoising autoencoder for pre-training sequence-to-sequence models. The model is composed of an encoder and a decoder. The encoder is a bidirectional encoder such as BERT, and the decoder is GPT, an autoregressive decoder. Different pre-training objectives were tested, such as token masking, token infilling, and sentence permutations. The effectiveness of such pre-training objectives depends on the end tasks. The BART-base model consists of 6 encoders and 6 decoders. It has 140M parameters. The BART-large model consists of 12 encoders and 12 decoders, and has 400M parameters.

### A.5 GPT-$N$ Models

GPT is a neural language model that is pre-trained to predict the next word given all the previous words. The model consists of a stack of transformer decoders. GPT exists in different versions: GPT-1 was the original model. GPT-2 is a GPT model with 1.5 billion parameters, while GPT-3 can have up to 175 billion parameters.

### A.6 T5

T5 is a text-to-text transfer transformer [Raffel et al., 2020b]. It uses a unified architecture that can be trained on a variety of NLP problems. Each problem is formulated as a text-to-text approach. It consists of an encoder-decoder architecture that is similar to the BERT model. However, T5 uses a causal self-attention and a fill-in-the-blank denoising pre-training objective. There are different T5 models with different sizes: The smallest version of T5 consists of 12 layers of transformer blocks with a hidden size of 512. It has 60M parameters. The largest T5 model consists of 24 layers of transformer blocks with a hidden size of 1024. It has 11B parameters.

## Appendix B. Datasets

### B.1 ParaRules

ParaRules [Clark et al., 2020] is a dataset that serves to evaluate deductive reasoning capabilities in language models. It consists of 40K synthetic questions. These instances were generated for 2K paraphrased facts, which were acquired by crowdworkers. Here is an example:

|  |  |
|---:|:---|
| **Context:** | Harry can do magic. |
|  | Muggles cannot do magic. |
|  | If a person can do magic then they can vanish. |
|  | Mr Dursley is a Muggle. |
| **Question:** | Can Harry vanish ? |
| **Expected answer:** | True |

## B.2  ProtoQA

ProtoQA [Boratko et al., 2020] is a question-answer dataset that is designed to evaluate commonsense reasoning capabilities in prototypical situations. A prototypical situation is represented as a question that can have multiple common answers. Here is an example with its possible answers:

|  |  |
|---:|:---|
| **Question:** | Name a profession where you might be fired if you lost your voice |
| **Expected answers:** | Radio host, Teacher |

The dataset is split into 9762 questions for training, 52 for validation, and 102 for testing.

## B.3  COM2SENSE

The COM2SENSE dataset [Singh et al., 2021] was designed to evaluate the commonsense reasoning capabilities in language models. The dataset includes 4K natural language true/false statements, with each sample paired with its complementary counterpart. The task consists of asking a model to judge whether a given sentence is logically coherent or not:

|  |  |
|---:|:---|
| **Context:** | Expecting ten fish in the net, Sammy was thrilled to see forty fish swimming in there. |
| **Expected answer:** | Coherent |

The authors created a counterpart to this question by modifying a few words:

|  |  |
|---:|:---|
| **Context:** | Expecting ten fish in the net, Sammy was thrilled to see *five* fish swimming in there. |
| **Expected answer:** | Incoherent |

## B.4  CODAH

The CODAH dataset [Chen et al., 2019] was designed to target the weaknesses of the state-of-the-art language models. The dataset was adversarially-constructed by allowing crowd workers to receive feedback from a pre-trained model and use this information to create challenging commonsense questions. The dataset consists of 2801 questions. The following is an example from the dataset:

|  |  |
|---:|:---|
| **Context:** | A man on his first date wanted to break the ice. He |
| **Choices:** | (A) drank all of his water. |
|  | (B) threw the ice at the wall. |
|  | (C) looked at the menu. |
|  | **(D) made a corny joke.** |

## B.5 CATS

The CATs dataset [Zhou et al., 2020b] reframes 6 different commonsense reasoning benchmarks to evaluate pre-trained transformer-based models on word-level and sentence-level tasks. These 6 different benchmarks are Sense Making [Wang et al., 2019], the Winograd Schema Challenge [Levesque et al., 2012], SWAG [Zellers et al., 2018], HellaSwag [Zellers et al., 2019], Sense Making with Reasoning [Wang et al., 2019], and the Argument Reasoning Comprehension Task [Habernal et al., 2018]. Also, they created a new task called Conjunction Acceptability to evaluate logical commonsense-knowledge in language models. Here is an example from CATs:

**Choices:** (A) Money can be used for buying **cars**.
(B) Money can be used for buying **stars**.
**Expected Answer:** (A)

Here, the model has to differentiate between statements that make sense and statements that don't.

## B.6 PIQA

The PIQA dataset [Bisk et al., 2020] is a benchmark to evaluate the physical commonsense capabilities of language models. It consists of a set of questions, where each question has two possible answers, but only one is correct. The training set has around 16000 instances, while the validation set and the testing sets have around 2000 and 3000 examples, respectively. The following is an instance of the dataset:

**Context:** To make a hard shelled taco,
**Choices:** (A) put seasoned beef, cheese, and lettuce onto the hard shell.
**(B) put seasoned beef, cheese, and lettuce into the hard shell.**

## B.7 TIMEDIAL

TIMEDIAL [Qin et al., 2021] is a dataset to test temporal commonsense reasoning in dialogs. It consists of 1.1K dialogs represented as multiple-choice cloze tasks. This task requires deep reasoning capabilities, such as performing different arithmetic operations over temporal expressions with a need for commonsense reasoning. Here is an example:

**Context:** A: How long do you want the house? All summer ?
B: No, just for six weeks.
A: I'm afraid I can only rent it for two months.
B: My holiday is only, [MASK] but I think my brother
and his family would take it for the other two weeks .
**Choices:** (A) six decades
**(B) 45 days**
**(C) six weeks**
(D) two months

### B.8 TORQUE

The TORQUE dataset [Ning et al., 2020] is a reading comprehension dataset concerning temporal ordering. It consists of 21K questions, split into 80% for training, 5% for validation, and 15% for testing. Here is an example:

| | |
|---:|:---|
| **Context:** | Heavy snow is causing disruption to transport across the UK, |
| | with heavy rainfall bringing flooding to the south-west of England. |
| | Rescuers searching for a woman trapped in a landslide at her home |
| | in Looe, Cornwall, said that had found a body. |
| **Question:** | What events have already finished? |
| **Expected answers:** | searching, trapped, landslide, said, found |

### B.9 MCTACO

The MCTACO dataset [Zhou et al., 2019] was designed to evaluate temporal commonsense in transformer-based models. The dataset consists of 13K questions, split into 30% for development and 70% for testing. Here is an example:

| | |
|---:|:---|
| **Context:** | Mr. Barco has refused US troops or advisers but has accepted US military aid. |
| **Question:** | What happened after Mr. Barco accepted the military aid? |
| **Choices:** | (A) The aid was denied |
| | **(B) He received the aid** |
| | **(C) Things started to progress** |

In the above example, two answers are correct to the same questions.

### B.10 TRACIE

TRACIE [Zhou et al., 2021] is a temporal reasoning textual entailment dataset. It consists of 5.5K instances, split into 20% for training and 80% for testing. Each instance has a hypothesis that is querying either about the start time of an event or about the end time of an event. Here is an example:

| | |
|---:|:---|
| **Premise:** | Tom needed to get braces. He was afraid of them. |
| | The dentist assured him everything would be fine. |
| | Tom had them on for awhile. Once removed he felt it was worth it. |
| **Hypothesis:** | Tom avoids foods he can't eat with braces |
| | starts before the braces are removed. |
| **Expected answer:** | Entailment |

### B.11 RICA

RICA [Zhou et al., 2020a] is a dataset of cloze questions that can be used to assess common-sense reasoning capabilities. To build this dataset, the authors first created commonsense axioms such as "Larger objects can contain smaller objects" and then translated them into commonsense statements. RICA consists of 16000 commonsense statements, split into 80% for training, 10% for validation, and 10% for testing. The task is to guess the comparator, which is masked in the input sentence, as here:

|  |  |
|---:|---|
| **Context:** | A prindag is smaller than a flurberg, so a flurberg |
|  | is [MASK] likely to contains a prindag. |
| **Expected answer:** | more |

### B.12 LogiQA

LogiQA [Liu et al., 2020b] is a multiple-choice machine reading comprehension dataset. This task assesses the logical deductive ability of language models for the case where the correct answer to a question is not explicitly included in the passage. The corpus includes 8678 paragraph-question pairs translated from the National Civil Servants Examination of China. Each question has one correct answer from a choice of four possible answers, as here:

|  |  |
|---:|---|
| **Context:** | David knows Mr. Zhang's friend Jack, and Jack knows David's friend Ms. Lin. |
|  | Everyone of them who knows Jack has a master's degree, |
|  | and everyone of them who knows Ms. Lin is from Shanghai. |
| **Question:** | Who is from Shanghai and has a master's degree? |
| **Choices:** | (A) David (B) Jack (C) Mr. Zhang (D) Ms. Lin |

The dataset is split into 80% for training, 10% for validation, and 10% for testing.

### B.13 ReCLOR

ReCLOR [Yu et al., 2020] is a multiple-choice machine reading comprehension dataset that tests logical reasoning. The corpus consists of questions retrieved from standardized exams such as LSAT and GMAT. It consists of 6138 paragraph-question pairs. Here is an example:

**Context:** Heavy rains during Centralia's corn planting season prevented some farmers there from planting corn. It is now the planting season for soybeans, another of Centralia's principal crops, and those fields originally intended for corn are dry enough for planting. Nonetheless, even though soybean prices are unusually high at present, the farmers will leave most of these fields empty rather than plant them with soybeans, since

**Question:** Which of the following most logically completes the passage below ?

**Choices:** (A) some Centralian farmers anticipate serious financial losses due to the extremely wet spring planting season.
(B) the extensive rains have led to an increase in the price of corn.
**(C) chemicals that were used to prepare the fields for corn planting would stunt the growth of soybeans.**
(D) many centralian farmers grow both corn and soybeans.

To adequately evaluate a model without allowing it to take advantage of artifacts in the corpus, the authors split the testing set into two sets: the EASY set where the instances are biased and the HARD set where they are not.

## B.14 AR-LSAT

AR-LSAT [Zhong et al., 2021] is a machine reading comprehension dataset that can be used to evaluate logical reasoning capabilities. The dataset was constructed by selecting the analytical reasoning section of 90 LSAT exams from 1991 to 2016. It consists of 2046 multiple-choice questions. Here is an example:

**Context:** A professor must determine the order in which five of her students
— Fernando, Ginny, Hakim, Juanita, and Kevin —
will perform in an upcoming piano recital.
Each student performs one piece, and no two performances overlap.
The following constraints apply:
Ginny must perform earlier than Fernando.
Kevin must perform earlier than Hakim and Juanita.
Hakim must perform either immediately before or immediately after Fernando

**Question:** If Juanita performs earlier than Ginny, then which one of the following could be true?

**Choices:** (A) Fernando performs fourth.
**(B) Ginny performs second.**
(C) Hakim performs third.
(D) Juanita performs third.
(E) Kevin performs second.

## B.15 QuAIL

QuAIL [Rogers et al., 2020a] is a machine reading comprehension dataset. It assesses verbal reasoning capabilities across 4 different domains: fiction, news, blogs, and user stories. The

corpus consists of 15K questions for 800 passages. The testing dataset comprises 15% of the questions, and different approaches were evaluated. Due to the size of the passages, we cannot show an example here.

### B.16 StrategyQA

StrategyQA [Geva et al., 2021] is a boolean QA benchmark that can be used to evaluate a model's reasoning capabilities. The model has to perform implicit decomposition of the question into reasoning steps in order to answer a question correctly. Here is an example:

|  |  |
|---:|:---|
| **Question:** | Did Aristotle use a laptop? |
| **Implicit Reasoning Steps:** | 1. When did Aristotle live? |
|  | 2. When was the laptop invented? |
|  | 3. Is #2 before #1? |
| **Expected answers:** | No |

The dataset is composed of 2780 instances, where each instance consists of a strategy question, a decomposition into reasoning steps, and Wikipedia paragraphs that answer each reasoning step.

### B.17 ConTROL

ConTRoL [Liu et al., 2020a] is a dataset of 8325 context-hypothesis pairs to evaluate a models' contextual reasoning capabilities over long texts. It is a passage-level textual entailment task that consists of context-hypothesis pairs. Here is an example:

|  |  |
|---:|:---|
| **Premise:** | Ten new television shows appeared during the month of September. |
|  | Five of the shows were sitcoms, three were hour- |
|  | long dramas, and two were news-magazine shows. |
|  | By January, only seven of these new shows were still on the air. |
|  | Five of the shows that remained were sitcoms. |
| **Hypothesis:** | At least one of the shows that were cancelled was an hour-long drama. |
| **Expected answer:** | Entailment |

### B.18 CLUTRR

CLUTRR [Sinha et al., 2019] is a benchmark dataset to evaluate the inductive reasoning capabilities of models. The task requires a model to infer the kinship between characters in short stories. Here is an example:

|  |  |
|---:|:---|
| **Context:** | Kristin and her son Justin went to visit her mother Carol |
|  | on a nice Sunday afternoon. They went out for a movie |
|  | together and had a good time. |
| **Question:** | How is Carol related to Justin ? |
| **Expected answer:** | Carol is the grandmother of Justin. |

CLUTRR is a synthetic dataset. For each experiment, 5000 instances were generated for training and 100 for testing.

### B.19 SVAMP

SVAMP [Patel et al., 2021] is a dataset that was created by varying instances of ASDiv-A (a dataset of one-unknown arithmetics problems). It contains 1000 tasks. To solve these tasks, a model needs a certain level of reasoning capability. It also has to be sensitive to the question. Here is an example:

|  |  |
|---:|:---|
| **Context:** | Jack had 8 pens and Mary had 5 pens. |
|  | Mary gave 3 pens to Jack. |
| **Question:** | How many pens does Jack have now? |
| **Expected answer:** | $8 + 3 = 11$ |

### B.20 MATH

MATH [Hendrycks et al., 2021] is a dataset that consists of 12500 competition mathematics problems. It is split into 7500 problems for training and 5000 for testing. Each instance is a description of the problem with a question, the step-by-step solution, and the final answer. Here is an example from the dataset:

|  |  |
|---:|:---|
| **Context:** | Tom has a red marble, a green marble, a blue marble, |
|  | and three identical yellow marbles. |
|  | How many different groups of two marbles can Tom choose? |
| **Expected answer:** | There are two cases here: |
|  | either Tom chooses two yellow marbles (1 result), |
|  | or he chooses two marbles of different colors ($\binom{4}{2}= 6$ results). |
|  | The total number of distinct pairs of marbles Tom can choose is |
|  | $1 + 6 = 7$ |

### B.21 IsarSTEP

IsarStep [Li et al., 2021] is a mathematical reasoning benchmark. It was built by collecting formal proofs written in Isabelle from the Archive of Formal Proofs and from the standard library of Isabelle/HOL. In this task a model needs to predict the missing intermediate proposition in a proof. Here is an example for the proof that $\sqrt{2}$ is not a rational number, where the missing intermediate proposition is *a is even*:

|  |  |
|---:|:---|
| **Context:** | $2b^2 = a^2 \Rightarrow [Missing\ Proposition] \Rightarrow \exists\ c \in\ \mathbb{Z}.\ a = 2c$ |
| **Expected answer:** | a is even |

The dataset is split into 820K examples for training, 5000 for validation, and 5000 for testing.

### B.22 HOList

HOList [Bansal et al., 2019] is an automated theorem proving dataset for higher-order logic. The benchmark includes 29465 theorems and their proofs, split into 60% for training, 20% for validation, and 20% for testing. Two tasks can be evaluated in HOList: (1) proving each theorem in the dataset, and (2) predicting the tactic and the arguments of the tactic that were used in the human proof. A tactic can be a previously proven theorem or a list of previously proven theorems.

### B.23 MetaMathStep

MetaMathStep [Polu and Sutskever, 2020] is a benchmark for automated theorem proving. The dataset evaluates the capabilities of a language model to generate a proof for a given statement. The dataset contains 3 million proof steps for around 38000 theorems, which are split into 36K for training, 1K for validation, and 1K for testing.

## Appendix C. Model Performances on Selected Datasets

| Type of Reasoning | Dataset | Model | Performance | Metric |
|---|---|---|---|---|
| Horn Reasoning (Section: 3.1) | ParaRules | RoBERTa | 98.8% | Accuracy |
| | ParaRules | PROVER (Based on RoBERTa) | 98.4% | Accuracy |
| | ParaRules | PRoofWriter (Based on T5) | 99.1% | Accuracy |
| Commonsense Reasoning (Section: 3.2) | ProtoQA | GPT-2 | 71.2% | Accuracy |
| | CODAH | BERT | 69.5% | Accuracy |
| | CATS | RoBERTa | 67% | Accuracy |
| | PIQA | RoBERTa | 77% | Accuracy |
| Event-based Commonsense Reasoning (Section: 3.3) | MCTACO | BERT | 69.9% | F1-Score |
| | TRACIE | SymTime (Based on T5) | 80.6% | F1-Score |
| | TORQUE | RoBERTa | 75.2% | F1-Score |
| Implicit Reasoning (Section: 3.4) | LogiQA | RoBERTa | 35.31% | Accuracy |
| | LogiQA | DAGN | 39.32% | Accuracy |
| | LogiQA | FOCAL REASONER (Based on RoBERTa) | 40.25% | Accuracy |
| | ReClor EASY | RoBERTa | 75.50% | Accuracy |
| | ReClor HARD | RoBERTa | 54.3% | Accuracy |
| | ReCLor EASY | DAGN (Based on RoBERTa) | 75.91% | Accuracy |
| | ReClor HARD | DAGN (Based on RoBERTa) | 44.46% | Accuracy |
| | ReCLor EASY | FOCAL REASONER (Based on RoBERTa) | 77.05% | Accuracy |
| | ReClor HARD | FOCAL REASONER (Based onRoBERTa) | 44.64% | Accuracy |
| | ReClor EASY | RoBERTa with Logical Data Augmentation | 81.4% | Accuracy |
| | ReClor HARD | RoBERTa with Logical Data Augmentation | 62.5% | Accuracy |
| | AR-LSAT | RoBERTa | 23.1% | Accuracy |
| | QuAIL | BERT | 55.9% | Accuracy |
| | StrategyQA | RoBERTa | 66% | Accuracy |
| | ConTRoL | BART | 60.95% | Accuracy |
| | CLUTTR | GAT | 77% | Accuracy |
| | RICA | GPT-2 | 50% | Accuracy |
| | RICA | RoBERTa | 50% | Accuracy |
| Mathematical Reasoning (Section: 3.5) | SVAMP | RoBERTa Embeddings + Graph2Tree | 65% | Accuracy |
| | MATH | GPT-2 | 6.9% | Accuracy |
| | IsarStep | Hierarchical Transformer | 22.8% | Accuracy |

Table 1: Model Performances on Selected Datasets

