# OpenReview forum: "Reasoning with Transformer-based Models: Deep Learning, but Shallow Reasoning"
_AKBC.ws/2021/Conference — AKBC 2021_

### Official Review · Reviewer_RsKf · 2021-07-07
**A good overview of how transformer models perform on logical reasoning tasks, though the scopes of "transformer models" and "logical reasoning" are not well-defined.**

**Rating:** 7
**Confidence:** 4

**Review:**

**Summary:** The paper surveys the ability of pretrained transformer-based models on logical reasoning tasks.

* Section 2 explores common reasoning abilities required to solve logical reasoning tasks.
* Section 3 considers different logical reasoning tasks, including both standard and adversarial variants. It also outlines the performance on such tasks of different model variants (i.e., pre-training only, fine-tuned on the target task, and fine-tuned with additional training objectives or symbolic components).
* Finally, Section 4 touches on tasks that are theoretically impossible for a transformer-based model to completely solve. It also presents natural language versions of such theoretical tasks.

**Strengths:**

The content of this survey is comprehensive and appropriate for the page limit. It gives a good overview of the major logical reasoning tasks and how well the models perform on them. Section 3.6 in particular provides a good summary: the models perform well when all information is explicit, perform reasonably on commonsense reasoning, and struggle on adversarial tasks but can be remedied by additional techniques.

**Weaknesses:**

1. The **scope of the models** explored is not well-defined.
    - "Transformer-based models" is a very wide scope. Even when considering only BERT-like models (BERT, RoBERTa, T5, GPT-3, etc.) without additional modifications, the model performance still depends on many factors such as the model architecture, model size, pre-training data, pre-training task (mask filling, generation, sentence-level objective), fine-tuning data, and fine-tuning methods.
    - Most works cited in this paper use BERT or its close variants such as RoBERTa. It is unclear if the results carry on the models trained on generative objectives such as T5 and GPT-3.
    - The paper often makes claims that can be considered too general (e.g., "Pretrained transformers-based models **cannot** differentiate between positive and negative statements" in Section 2.1, or "mispriming works only when ..." in Section 2.2) while they might only apply to certain model variants.
    - The paper also mixes the results of models with and without fine-tuning, especially in Section 2. While these can be discerned from careful reading, it would be great to make this more clear to prevent readers from drawing incorrect conclusions.

2. Likewise, the **scope of the tasks** is not well-defined.
    - It is not explained what "logical reasoning" means, and the paper contains many examples that are not "logical reasoning" in the traditional sense. Examples include having world knowledge (e.g., that the iPhone was produced by Apple, as mentioned in the introduction), having commonsense knowledge (e.g., that ravens can fly, as mentioned in Section 3.2), and doing arithmetic (e.g., adding two numbers in Section 3.5). While these skills are useful for some logical reasoning tasks, I wouldn't call them logical reasoning on their own.
    - Mispriming (Section 2.2) does not seem to be a building block for any logical reasoning task. Though one could argue that the ability to distinguish useful from distracting contexts in general is an essential building block.
    - "Text understanding" (Section 3.4) also sounds too broad and encompasses the other types of reasoning in prior sections.

3. (minor) The natural language version of impossible reasoning tasks (Sections 4.2 and 4.3) is not very informative.
    - BERT is known to not generalize well to long inputs, especially when it's longer than what the model have seen during training. When the model performs poorly on the proposed light switch and cake tasks, it is unclear whether it's due to the length problem above or the theoretical impossibility of the task.

**Other comments:**

1. Title: What does "shallow reasoning" mean? Some of the presented tasks, such as the Horn clause one, require quite deep reasoning.
2. Section 2: The term "building block" is a bit ambiguous. I thought it meant model building blocks (e.g., a self-attention layer) during the first read. Maybe something like "building blocks for logical reasoning tasks", "subtasks", or "logical phenomena the models need to handle" would be clearer?
3. Section 2.4: "they will still reply" --- reply correctly or reply randomly?
4. Section 3.1: It would be nice to mention that all information is present in the task and no external knowledge is needed.
5. Section 3.2: While the listing of datasets is helpful as references, it would be nice to give some descriptions of what each dataset aims to do, possibly in the appendix. This comment also applies to other sections.
6. Section 3.4: "e.g.," --> "for example" (should be spelled out here)
7. Section 3.4: The provided example from LogiQA looks similar to the Horn clause examples in Section 3.1, yet the models struggle here. Why is this the case?

---

> ### Author Response · Authors · 2021-07-31
> **Thank you for your review and valuable feedback! We have uploaded an updated version of the paper with your suggestions.**
>
> We would like to thank you for your insightful and positive feedback. We are happy to learn that you found our survey comprehensive, and that it gives a good overview of the major reasoning tasks. We reply here in detail to your suggestions:
>
> **1. The scope of the models explored is not well-defined.**
> * That is true. We report on the performance of the best known models, as put forward by the cited works. Thus, our survey traces the current state of the art, as a lower bound of what is possible. We also briefly explain the architectures that were used to achieve these performances, but we cannot describe their entire configuration. For this, the reader is invited to read the respective papers. However, we now make explicit whether a model was finetuned or not.
> * This is true. We always cite the current best performing model. Unfortunately, we cannot establish whether another model would have had a better performance. This is something we leave to the individual papers that we cite. Our survey allows the reader to understand the current state of the model performance.
> * That is indeed true for Section 2. We have now made it clear that these comments apply mainly to BERT.
> * Indeed. We now specify for each model whether it was finetuned or not.
>
> **2. Likewise, the scope of the tasks is not well-defined**
> * Yes, we agree. We have replaced “logical reasoning” by “reasoning” because as you said not all the tasks fall under “logical reasoning.”
> * Exactly, the ability to distinguish useful from distracting contexts in general is an essential building block. Thank you for proposing this reformulation! We now use it in the paper.
> * Yes, we agree. In the updated version of the paper, we replaced “Text understanding” with “Implicit Reasoning”. The datasets in this section cover different logical reasoning types such as deductive, inductive and abductive reasoning. Different from the other subsections, they do not explicitly specify the rules, and they rely less on common sense knowledge. We have now made that explicit.
>
> **3. (minor) The natural language version of impossible reasoning tasks (Sections 4.2 and 4.3) is not very informative.**
> * There are some tasks where transformer-based models can generalize to longer inputs than those seen during training. The work “On the Ability and Limitations of Transformers to Recognize Formal Language” shows different such tasks (https://arxiv.org/pdf/2009.11264.pdf).
> For the specific tasks of Dyck and Parity, however, we know that transformer-based models cannot generalize due to principal limitations. This is what our experiments also show in practice. Just for curiosity, we ran an experiment where we trained the Light-Switch-Task on sentences with 16-20 switch operations, and tested on sentences with 1-15 switch operations. The results are still dismal, as expected, with a precision of 50%.
>
> **Other comments:**
>
> **1. Title: What does "shallow reasoning" mean? Some of the presented tasks, such as the Horn clause one, require quite deep reasoning.**
> * We modified the title to “Reasoning with Transformer-based Models: Deep learning, _but_ Shallow Reasoning”, to point out that transformers in general cannot solve reasoning tasks in a way a human can. This does not exclude that they can solve some tasks of impressive complexity, including the Horn rule reasoning that you mention.
>
> **2. Section 2: The term "building block" is a bit ambiguous. I thought it meant model building blocks (e.g., a self-attention layer) during the first read. Maybe something like "building blocks for logical reasoning tasks", "subtasks", or "logical phenomena the models need to handle" would be clearer?**
> * Thank you! We updated the section title to “Common Pitfalls for BERT-like Models”.
>
> **3. Section 2.4: "they will still reply" --- reply correctly or reply randomly?**
> * In the example given in Section 2.4, “still reply” means “still reply correctly”. We have fixed this. Thank you!
>
> **4. Section 3.1: It would be nice to mention that all information is present in the task and no external knowledge is needed.**
> * Thank you! We added this observation to Section 3.1.
>
> **5. Section 3.2: While the listing of datasets is helpful as references, it would be nice to give some descriptions of what each dataset aims to do, possibly in the appendix. This comment also applies to other sections.**
> * Following your suggestion, we added an extensive appendix that describes the challenging datasets.
>
> **6. Section 3.4: "e.g.," --> "for example" (should be spelled out here)**
> * Thank you, we fixed it.
>
> **7. Section 3.4: The provided example from LogiQA looks similar to the Horn clause examples in Section 3.1, yet the models struggle here. Why is this the case?**
> * In contrast to the Horn clauses examples in Section 3.1, the examples in Section 3.4 do not explicitly provide the facts and rules to the model. Hence, the model needs to perform implicit reasoning to solve such tasks. We have now clarified this in the paper.

---

### Official Review · Reviewer_Nrsx · 2021-07-19
**Survey paper on recent work applying transformers for a variety of reasoning tasks.**

**Rating:** 6
**Confidence:** 4

**Review:**

Overview:
This paper provides a survey on recent work in the area of reasoning using transformer based models. It first describes common issues that are faced by transformers such as falling prey to lexical overlap and indifference to word order, and then goes on to describe different kinds of reasoning that can be tackled using transformers and the relevant work tackling each. Finally, the paper gives two examples of types of tasks that are beyond the capabilities of transformers.

High level comments:
The paper is well written, easy to read and covers many sub-types of reasoning tasks. It would benefit from being more detailed on the models that are being evaluated on each of the datasets, maybe in some form of aggregated table so the reader has a sense of which models perform better on which types of reasoning tasks. Since the different models - GPT-3/BERT/etc. are all trained on different amounts of data as well as different types of data, it would be very useful to know what scale of data and size of model is required for good performance on these various tasks. I believe this would be a more useful addition to the paper than section 3.5.

Detailed comments:
I would rename the first section as “basic building blocks” doesn’t seem appropriate when one of the subtitles is “mispriming”. This section seems to cover the artifacts that the model needs to learn to be resilient to, such as not falling prey to mispriming, not overly relying on pattern heuristics, not only taking into account words without using order information etc. Maybe a better section title would be “Commonly observed pitfalls and ways to avoid them.” ?
Similarly, I feel the section 2 title would be more informative if you named it “Types of reasoning with transformers” since just “Reasoning with transformers” is not very informative given that it’s the topic of the whole paper.

Section 3.5, It would be useful to the reader if you give some intuitions on *why* it’s not possible to solve tasks such as parity using transformers in addition to stating that a certain paper shows that it cannot. For instance, in parity tasks, it requires the model to aggregate information from each segment and since all the tokens are processed in parallel by self attention, it would need the network depth to increase proportionally to the length of the sequence, to be able to solve the task making it not possible for non-recurrent models like transformers solve the task.

In the lights example, it is mentioned that the model is trained for 50 iterations. Did the model learn the task in these 50 iterations? Did the training converge? It’s hard to make conclusions about why the test precision was 0.5 - whether it is because the model was unable to learn the task during training or if it has to do with the train test distribution shift which causes the model to not do well on the task, without this information.

Same issue with the cake task: missing details about model convergence and performance on the training data. Further it would be interesting to know if changing the type of positional encodings (learnt embeddings/sinusoidal embeddings/CAPE [1] ) can change the results for generalizing to longer length.

Missing citation:
Transformers for symbolic math : Lample, Guillaume, and François Charton. "Deep learning for symbolic mathematics.", ICLR 2020.

Minor typos:
Intro last paragraph, make transformer-based reason -> make transformer-based models reason
Section 3.4 Did you mean to say performer based models?
Section 4.2 contining -> containing

[1] CAPE: Encoding Relative Positions with Continuous Augmented Positional Embeddings

---

> ### Author Response · Authors · 2021-07-31
> **Thank you for your review and valuable feedback! We have uploaded an updated version of the paper with your suggestions.**
>
> We would like to thank you for your review, and for your positive assessment of our paper! We are happy that you found the paper well written, easy to read, and extensive. We reply to your questions below.
>
> **1. High level comments: The paper is well written, easy to read and covers many sub-types of reasoning tasks. It would benefit from being more detailed on the models that are being evaluated on each of the datasets, maybe in some form of aggregated table so the reader has a sense of which models perform better on which types of reasoning tasks. Since the different models - GPT-3/BERT/etc. are all trained on different amounts of data as well as different types of data, it would be very useful to know what scale of data and size of model is required for good performance on these various tasks. I believe this would be a more useful addition to the paper than section 3.5.**
>
> Thank you for this suggestion! We have now included an appendix in the paper, which describes the basic models, the challenging datasets, and the performance of the models.
>
> **2. Detailed comments: I would rename the first section as “basic building blocks” doesn’t seem appropriate when one of the subtitles is “mispriming”. This section seems to cover the artifacts that the model needs to learn to be resilient to, such as not falling prey to mispriming, not overly relying on pattern heuristics, not only taking into account words without using order information etc. Maybe a better section title would be “Commonly observed pitfalls and ways to avoid them.” ? Similarly, I feel the section 2 title would be more informative if you named it “Types of reasoning with transformers” since just “Reasoning with transformers” is not very informative given that it’s the topic of the whole paper.**
>
> Yes we agree. We have replaced  ”basic building blocks” by “common pitfalls for BERT-like models”, as you suggested, and we have replaced the title of Section 2 by “Types of reasoning with transformers-based models.”
>
> **3. Section 3.5, It would be useful to the reader if you give some intuitions on why it’s not possible to solve tasks such as parity using transformers in addition to stating that a certain paper shows that it cannot. For instance, in parity tasks, it requires the model to aggregate information from each segment and since all the tokens are processed in parallel by self attention, it would need the network depth to increase proportionally to the length of the sequence, to be able to solve the task making it not possible for non-recurrent models like transformers solve the task.**
>
> Correct. Thank you for suggesting these intuitive explanations! We have added them to Section 4.1!
>
> **4. Same issue with the cake task: missing details about model convergence and performance on the training data.**
>
> We have added the performance of the models on the training and validation sets. Each time, the models then achieve an F-score > 0.99.
>
> **Further it would be interesting to know if changing the type of positional encodings (learnt embeddings/sinusoidal embeddings/CAPE [1] ) can change the results for generalizing to longer length.**
>
> This would indeed be interesting to know! Unfortunately, we could not verify this in the frame of the rebuttal period. But we have added this idea in the conclusion.
>
> **5. Missing citation: Transformers for symbolic math : Lample, Guillaume, and François Charton. "Deep learning for symbolic mathematics.", ICLR 2020.**
>
> Thank you, we added this work to Section 3.5 (Mathematical Reasoning Tasks).
>
> **6. Minor typos: Intro last paragraph, make transformer-based reason -> make transformer-based models reason Section 3.4 Did you mean to say performer based models? Section 4.2 contining -> containing**
>
> Thank you for spotting these typos! We fixed them. In Section 3.4, we meant indeed “transformer-based models.”

---

### Official Review · Reviewer_6i26 · 2021-07-23

**Rating:** 6
**Confidence:** 3

**Review:**

### Summary

This paper surveys the performance of BERT on tasks that requires logical reasoning. The survey first introduces basic building blocks that a model should have in order to perform reasoning on natural language (e.g. handling negations, stability to word-order, etc). Next it describes a series of reasoning tasks and report performance on BERT-based models. BERT-based model performs well on reasoning tasks when all information required to perform deductive reasoning, such as facts and rules, are provided (e.g. logical reasoning). However, when certain information needs to be inferred (background knowledge), these models fail (text understanding, mathematical reasoning). Finally, the survey discusses two tasks (Parity and Dyck-2) that has been theoretically proved that transformers cannot perform and the paper comes up with two example reasoning tasks and shows that transformer models indeed fail on them.

### Strengths:

1. The paper presents a comprehensive list of reasoning tasks and cites many relevant work and can serve as a go-to survey for the BERT-based model's performance on natural language reasoning tasks

### Weaknesses:

1. The survey currently reads like a list of reasoning tasks which BERT mostly fail at and one or two example, demonstrating where it failed. It would be more helpful to have a discussion around each of the failed reasoning tasks. For example, why is BERT not able to perform those reasoning tasks? What % of examples need this reasoning, does the model fail at? Do they fail at all examples?
2. I think the paper would benefit if it clearly lays down what the audience will learn after reading the survey. A discussion about how we can make the transformer model perform reasoning could also be helpful

Missing citations:
Boratko et al 2020 ProtoQA: a question answering dataset for prototypical common-sense reasoning shows another case where pre-trained LMs don't perform well because of the lack of background knowledge.

---

> ### Author Response · Authors · 2021-07-31
> **Thank you for your review and valuable feedback! We have uploaded an updated version of the paper with your suggestions.**
>
> We are happy to see that you found our list of reasoning tasks comprehensive, our citations extensive, and the survey in general to be a reference for the performance of transformer-based models on natural language reasoning tasks! We reply to your suggestions below.
>
> **1. The survey currently reads like a list of reasoning tasks which BERT mostly fail at and one or two example, demonstrating where it failed. It would be more helpful to have a discussion around each of the failed reasoning tasks. For example, why is BERT not able to perform those reasoning tasks? What % of examples need this reasoning, does the model fail at? Do they fail at all examples?**
>
> We agree that such an analysis is of utmost interest. To some degree, the individual articles that we cite actually provide such an analysis. Our article, in contrast, takes a breadth-first approach: It provides an overview of the tasks, categorizes them, and describes the approaches that have been used to tackle them. It thus serves as an entry point to the cited papers, rather than as a discussion of individual performances. A general interpretation of why and where transformers fail is in Section 3.6.
>
> **2. I think the paper would benefit if it clearly lays down what the audience will learn after reading the survey. A discussion about how we can make transformer model perform reasoning could also be helpful.**
>
> Our article traces the boundary of the reasoning that is currently possible with transformer models. Our paper thus serves two purposes: First, the audience is made aware of the weaknesses of current models, so as to sensibilize them to pitfalls they may encounter in applications. Second, the audience can use our work to find interesting research questions that are still open. As you requested, we also provide an outlook of possible research avenues in the last paragraph of Section 3.6.
>
> **3. Missing citations: Boratko et al 2020 ProtoQA: a question answering dataset for prototypical common-sense reasoning shows another case where pre-trained LMs don't perform well because of the lack of background knowledge.**
>
> Thank you! We added this reference in Section 3.2.

---

### Official Review · Reviewer_8qLK · 2021-07-23
**Nice survey on some of the limitations of modern transformer-based models**

**Rating:** 6
**Confidence:** 4

**Review:**

Authors survey the literature on probing the understanding, reasoning, and commonsense properties of transformer-based models, outlining several limitations of current models. They cover mispriming, negations, the tendency of such models to learn simple heuristics, under-sensitivity to word order, reasoning with patterns resembling Horn clauses, commonsense reasoning, reasoning with temporal knowledge. Furthermore, they also cover several natural language understanding tasks such as natural language inference and mathematical reasoning.

It would be nice to have snippets from the different datasets, to get a better understanding of what's out there for training and evaluating our models, but I understand the space limitations.

There are a few missing references:
- On the Power of Saturated Transformers: A View from Circuit Complexity (https://arxiv.org/abs/2106.16213)
- A couple of systems focus on solving CLUTRR using dedicated architectures: https://arxiv.org/abs/2007.06477 and https://github.com/ML-KULeuven/deepproblog

---

> ### Author Response · Authors · 2021-07-31
> **Thank you for your review and valuable feedback! We have uploaded an updated version of the paper with your suggestions.**
>
> We thank you for your feedback! We are glad that you found our survey useful! Thank you also for providing us with the missing references!
>
> **1. It would be nice to have snippets from the different datasets, to get a better understanding of what's out there for training and evaluating our models, but I understand the space limitations.**
>
> Thank you for your suggestion. We have now added an appendix to the paper, which   includes the descriptions of the basic models, a description of the challenging datasets, and a table of the model performances. The appendix will be part of the Arxiv version of our paper.
>
>
> **2.  There are a few missing references:**
>
> **On the Power of Saturated Transformers: A View from Circuit Complexity (https://arxiv.org/abs/2106.16213)**
>
> Thank you for this reference! We have added in the conclusion a proposal for future work, which is to test the saturated attention mechanism on our natural tasks (light switch and cake tasks).
>
> **A couple of systems focus on solving CLUTRR using dedicated architectures: https://arxiv.org/abs/2007.06477 and https://github.com/ML-KULeuven/deepproblog**
>
> Thank you, we have added these references in Section 3.4 as neuro-symbolic   approaches!

---

### Author Response · Authors · 2021-07-31
**Thank you for your review and valuable feedback! We have uploaded an updated version of the paper with your suggestions.**

We would like to thank all the reviewers for their valuable feedback! We are happy that all reviewers appreciated our survey paper!

Our article traces the boundary of the reasoning that is possible with transformer-based models in the current state of the art. It takes a breadth-first approach: It provides an overview of reasoning tasks, categorizes them, and describes the approaches that have been used to tackle them. It thus serves two purposes: First, the audience is made aware of the weaknesses of current models, so as to sensibilize them to pitfalls they may encounter in applications. Second, the audience can use our work to find interesting research questions that are still open. Our work is also the first survey of such reasoning with transformer-based models.

A common request by the reviewers was to provide more details on the dataset and models. We have added an appendix to the paper, which includes the descriptions of the basic models, a description of the challenging datasets, and a table of the model performances. This appendix can be found in the updated version of the paper. Should the paper get accepted, the appendix will be published in the Arxiv version of the paper.

We also reply individually to each reviewer.

---

### Decision · Program_Chairs · 2021-08-18

**Decision:**

Accept

**Comment:**

The paper surveys the ability of pretrained transformer-based models on logical reasoning tasks. This is a blooming area of research which makes this paper particularly attractive to AKBC community. Authors introduce basic capabilities in order to do reasoning (e.g. handling negations, stability to word-order, etc). Later, they survey a series of reasoning tasks and report performance on BERT-based models. The main finding is that BERT-based model performs well when all information (rules, facts) required to perform deductive reasoning is given as input. When any of this information is missing, models fail to infer those which is necessary to be successful at reasoning. Reviewers have given excellent set of comments to improve the draft in terms of scope of models and tasks, and main takeaways for the reader. We urge authors to incorporate missing details in the next draft of this paper.